# Integrating Clinical and Imaging Markers for Survival Prediction in Advanced NSCLC Treated with EGFR-TKIs

**DOI:** 10.3390/cancers17152565

**Published:** 2025-08-03

**Authors:** Thanika Ketpueak, Phumiphat Losuriya, Thanat Kanthawang, Pakorn Prakaikietikul, Lalita Lumkul, Phichayut Phinyo, Pattraporn Tajarernmuang

**Affiliations:** 1Division of Oncology, Department of Internal Medicine, Faculty of Medicine, Chiang Mai University, Chiang Mai 50200, Thailand; thanika050@gmail.com; 2Division of Pulmonary, Critical Care and Allergy, Department of Internal Medicine, Faculty of Medicine, Chiang Mai University, Chiang Mai 50200, Thailand; theart9057@gmail.com; 3Department of Radiology, Faculty of Medicine, Chiang Mai University, Chiang Mai 50200, Thailand; thanat.kanthawang@cmu.ac.th (T.K.); pakorn.p@cmu.ac.th (P.P.); 4Center for Clinical Epidemiology and Clinical Statistics, Faculty of Medicine, Chiang Mai University, Chiang Mai 50200, Thailand; lalita.lumkul@gmail.com (L.L.); phichayut.phinyo@cmu.ac.th (P.P.); 5Center of Multidisciplinary Technology for Advanced Medicine (CMUTEAM), Faculty of Medicine, Chiang Mai University, Chiang Mai 50200, Thailand; 6Department of Biomedical Informatics and Clinical Epidemiology, Faculty of Medicine, Chiang Mai University, Chiang Mai 50200, Thailand

**Keywords:** non-small cell lung cancer, epidermal growth factor receptor, EGFR, outcome, predictive score

## Abstract

A high prevalence of epidermal growth factor receptor (EGFR) mutations in non-small cell lung cancer (NSCLC) has been observed among the Asian population, along with significantly lower survival rates reported in previous real-world studies compared to clinical trials. Therefore, a precise survival prediction model for this patient subgroup is crucial. A prognostic model was developed using fundamental clinical and non-cancerous radiologic parameters to predict 18-month mortality in patients with EGFR-mutated advanced NSCLC. Key factors included a BMI <18.5 or ≥23, presence of bone metastasis, neutrophil-to-lymphocyte ratio ≥ 5, albumin-to-globulin ratio < 1, and mean pulmonary artery diameter ≥ 29 mm. The model demonstrated good predictive accuracy and is available as a web-based tool for clinical use.

## 1. Introduction

Lung cancer is a leading cause of death accounting for approximately 1.79 million (18% of all cancer) deaths worldwide in 2020 [1]. In Thailand, it was the second most common cancer in the same year, with an estimated 23,713 new cases and 20,395 deaths [2]. Non-small cell lung cancer (NSCLC) is the predominant type, with adenocarcinoma being the more frequently observed histology subtype [3]. Molecular testing plays a critical role in NSCLC to guide prognosis and treatment strategies [4].

Among molecular alterations in NSCLC, epidermal growth factor receptor (EGFR) mutations are significant contributors to oncogenesis. The incidence of EGFR mutations varies geographically, but with a high prevalence of 45–50% among Asian populations with NSCLC [4]. The most common mutations include deletions in exon 19 and the L858R substitution mutation in exon 21 [5]. These mutations are strongly associated with a favorable response to tyrosine kinase inhibitors (TKIs), which have become the standard treatment for advanced NSCLC with EGFR mutations [3,6]. The survival benefits of TKIs depend on their generation. For example, the phase III FLAURA trial demonstrated a median overall survival (OS) benefit of 38.6 months with a first-line treatment with osimertinib, a third generation EGFR-TKI, compared to 31.8 months with a first-generation EGFR-TKI [7].

In Thailand, a retrospective multicenter study conducted by Sukauichai et al. (2013–2019) reported that the median OS for advanced NSCLC patients treated with EGFR-TKIs was significantly longer compared to those receiving chemotherapy alone (19.08–32.46 months vs. 11.10 months, *p* = 0.006) [8]. However, due to reimbursement constraints, most patients with advanced EGFR-mutated NSCLC in Thailand are initially treated with first-generation EGFR-TKIs.

Prognostic factors influencing survival in NSCLC have been widely studied. Gong J et al. identified poor performance status (PS), elevated lactate dehydrogenase (LDH) levels, thrombocytosis, and smoking history as independent predictors of poor prognosis in advanced NSCLC [9]. Ping Lu et al. demonstrated that a low serum albumin to globulin ratio (AGR < 1.12) was independently associated with shorter progression-free survival (PFS) and OS [10]. Similarly, Ogura Y et al. found that inflammatory markers, such as neutrophil-to-lymphocyte ratio (NLR), C-reactive protein (CRP)-to-albumin ratio, and advanced lung cancer inflammation index, were significantly correlated with survival outcomes in patients treated with chemoimmunotherapy [11]. For EGFR-mutated NSCLC, meta-analysis suggested that patient with exon 19 deletions had better PFS compared to those with the L858R mutation in exon 21 [12]. Another study highlighted that Eastern Cooperative Oncology Group Performance Status (ECOG-PS) and brain metastasis significantly influence OS and PFS [13].

Host factors also influence lung cancer prognosis. For instance, a systematic review and meta-analysis of 26 observational studies revealed that coexisting chronic obstructive pulmonary disease (COPD) was strongly associated with poorer OS and disease-free survival (DFS), regardless of tumor stage. The presence of emphysema in patients with lung cancer was linked to a worse OS (HR 1.66) [14]. Another meta-analysis demonstrated the association of sarcopenia and shorter OS in all stages of NSCLC [15]. Non-cancerous chest computed tomography (CT) findings such as high coronary artery calcium (CAC) scores, increased pulmonary artery-to-aorta (PA/Ao) ratios, and reduced thoracic skeletal muscle index, were also independent predictors of worse outcomes [16]. Several studies have demonstrated good predictive performance of advanced radiomics features in NSCLC [17,18]. However, application of such models remains limited in resource-constrained setting due to complex imaging analysis, which is less accessible compared to more fundamental clinical parameters.

Given the high prevalence and distinct characteristics of EGFR-mutated NSCLC, developing a precise survival prediction model for this patient subgroup is crucial. Although these patients generally exhibit favorable prognoses, some experience rapid progression and early mortality. The aim of this study was to identify clinical factors and non-cancerous findings from routine chest CT scans associated with early mortality and develop a prognostic prediction model in patients with advanced EGFR-mutated NSCLC.

## 2. Materials and Methods

### 2.1. Study Design, Participants, and Data Collection

This was a retrospective study that collected the data from Chiang Mai University Hospital between January 2012 and October 2022. This study included adult patients (age ≥ 18 years) diagnosed with advanced or locally advanced EGFR-mutated NSCLC confirmed by pathological evaluation. All patients received at least one dose of a first-line EGFR-TKI as a treatment for advanced or locally advanced NSCLC. The study population was identified using ICD-10 codes (C341–343, C348–349, D381) and prescription records for EGFR-TKIs, including gefitinib, erlotinib, afatinib, and osimertinib. Inclusion criteria required availability of pre-treatment laboratory data such as complete blood count (CBC), creatinine (Cr), liver function test (LFT), and chest CT scan. Patients were excluded if they lacked these pre-treatment data, were receiving palliative end-of-life care at initial diagnosis without EGFR-TKI therapy, or received EGFR-TKI as a subsequent line of treatment. The platelet-to-lymphocyte ratio (PLR), the neutrophil-to-lymphocyte ratio (NLR), and the albumin-to-globulin ratio (AGR) were calculated from the pre-treatment laboratory results.

Clinical characteristics, including age, gender, body mass index (BMI), type of surgical procedures or interventions, comorbidities, laboratory results, imaging findings, pathological reports, disease staging, admission date, and discharge date were extracted from electronic medical records. Chest CT scan parameters including presence of emphysema, size of main pulmonary artery (MPA) and aorta (AO) at bifurcation, and MPA to AO ratio were assessed. All parameters in CT were measured on Picture Archiving and Communication System (PACS) workstations by two independent radiologists with 3 and 8 years of experience (P.P. and T.K., respectively), who were blinded to clinical data and ensured unbiased assessment. This study was approved by research ethics committee of Faculty of Medicine, Chiang Mai University (Research ID: 9423, Study code MED 2566-09423).

### 2.2. Data Management and Confidentiality

Data were collected electronically and stored in a secure, encrypted database accessible only to the research team. Access to protected health information (PHI) was strictly limited to authors. Identification data were stored separately in an encrypted format using the hospital’s current IT system and housed on a protected-password internal server. To ensure confidentiality, patients were assigned unique identification numbers, and all identifiable information was replaced with random codes, effectively de-identifying the data. The dataset used for analysis was entirely de-identified, ensuring that no patient could be personally identified. These unique identifiers and random codes were used exclusively for statistical analysis. After the publication of this study, patient identities will remain confidential.

### 2.3. Statistical Analysis

Statistical analysis was performed using Stata version 16 (StataCorp, College Station, TX, USA) and R statistical software version 2024.4.2.764 (R Core Team, Vienna, Austria). The significance level was set at a p-value of less than 0.05. Sample size calculations, based on previous literature, were conducted using Stata with an alpha of 0.05 (two-sided test) and power of 0.80, yielding a minimum sample size of 186 cases (Appendix A). Continuous data were summarized using mean and standard deviation (SD) for normally distributed data and median and interquartile range (IQR) for non-normally distributed data. Counts and percentages were used for categorical data.

### 2.4. Study Variables and Predictors

Prognostic factors including clinical characteristics, pre-treatment laboratory results, and parameters associated with survival outcomes were explored. The selection of prognostic predictors was based on the availability of predictors at the time of prediction, a review of the existing literature, and clinical expertise. Continuous predictors were categorized at generally accepted cutoff points, according to those in the literature. Kaplan–Meier curves were used to illustrate survival differences based on prognostic factors. Differences in survival distribution after 6, 12, and 18 months across prognostic factors were examined using the log-rank test. Cox’s proportional hazard regression was used to estimate the hazard ratio (HR) for mortality. Significant variables (*p*-value < 0.05) from univariable analysis were included in the model.

### 2.5. Missing Data Handling

Missing values in prognostic factors were addressed through imputation using the k-Nearest Neighbor (kNN) imputation based on a variation of the Gower distance [19]. Only prognostic factors were included in imputation procedures. The imputation was conducted in R using a kNN function from the VIM package with default arguments set to 5 neighbors, the median function for numerical data, and the maximum function for categorical data. To assess the robustness of our imputation approach, we also performed sensitivity analyses using the complete case analysis approach.

### 2.6. Derivation of the Survival Model

The prognostic model was derived using a flexible parametric survival model (Royston–Parmar (RP) model) via the stpm2 package in Stata version 16 (StataCorp, College Station, TX, USA). The proportional hazard assumption was tested using Schoenfeld’s residuals before fitting the RP model. The cumulative hazard scale with three degrees of freedom was selected based on the lowest Akaike Information Criterion (AIC) and Bayesian Information Criterion (BIC) values. Significant variables from univariable analysis were included in the multivariable flexible parametric model, with pre-specified hazard scale and number of knots. Backward elimination was performed on each predictor using a significance threshold of *p*-value < 0.05.

### 2.7. Discrimination and Calibration

Model performance was evaluated based on discrimination and calibration. Discrimination was assessed using Harrell’s C-statistic to determine the model’s ability to distinguish between patients with longer and shorter survival times. Additional discrimination statistics, including Somers’ D coefficient for Harrell’s C, Royston and Sauerbrei’s D statistic, and R^2^_D_, were also reported. Calibration was assessed by comparing predicted survival probabilities with observed survival outcomes using a calibration plot. Survival probabilities were divided into four quantiles, creating four distinct risk groups. The calibration plot was then examined to compare the agreement between the model-predicted survival curves and the Kaplan–Meier survival curves within each quantile.

### 2.8. Internal Validation

Internal validation and assessment of model optimism were conducted using the bootstrap resampling method. Two hundred samples, each equal in size to the original dataset, were generated through random sampling with replacement. The entire modeling process was repeated for each bootstrap sample, generating 200 unique models. Harrell’s C-statistics were calculated and averaged across all models, obtaining the apparent performance. Then, each bootstrap model was applied to the original dataset and Harrell’s C-statistic was calculated again, resulting in a performance test. The difference between these performance averages provided an estimate of the model’s optimism. Additionally, we reported the optimism for other discrimination measures, including Somers’ D coefficient for Harrell’s C, Royston and Sauerbrei’s D statistic, R^2^_D_, and the shrinkage factor, which particularly important for external validation studies.

### 2.9. Model Presentation

For applicability, the model was developed into a web application using Streamlit, an open-source Python framework for interactive data applications. Model coefficients and baseline survival rates after 6, 12, and 18 months were extracted to calculate survival probabilities at each time point. The model and Python script are available at https://github.com/Lalita-LL/lung-ca-survival-Aug2024 (accessed on 2 September 2024). The script was deployed with inputs including time to prediction and all predictors from the final model.

## 3. Results

### 3.1. Patient Characteristics

A total of 216 patients met the eligible criteria for this study. However, 27 patients were excluded due to incomplete pre-treatment CT results, leaving 189 patients for analysis. The baseline characteristics of the study population are summarized in Table 1. The average age of patients was 64.81 years (SD ± 10.76), with 133 patients (70.37%) being over 60 years old. Approximately one-third of the cohort was male (65 patients, 34.39%). The mean BMI was 22.38 kg/m^2^ (SD ± 4.05), with 44.44% of the patients falling within the normal BMI range (18.5–22.9 kg/m^2^). Most patients were never-smokers (62.96%) and the most common comorbid conditions were hypertension (43.39%), followed by diabetes mellitus (16.40%) and other second primary cancers (4.76%).

The majority of patients received a first-generation EGFR-TKI as first-line treatment, with 48.15% receiving gefitinib and 42.86% receiving erlotinib. Laboratory investigations and CT findings are detailed in Table 2. Among the patients, 48.13% of patients had a PLR greater than 200, while 20.86% had an NLR greater than 5. Additionally, 31.79% of patients exhibited an AGR of less than 1. Enlargement of the MPA size, defined as a diameter of at least 29 mm, was observed in 28.04% of patients. The average MPA-to-AO ratio was 0.84 (SD ± 0.12). The mutational analysis identified exon 19 deletion as the most common subtype (63.49%), followed by exon 21 L858R substitution (28.57%). Other rarer mutations are described in Appendix A.

### 3.2. Survival Rate of Patients

At the 18-month follow-up, 84 patients (44.44%) had died (Figure 1). The overall survival rates after 6, 12, and 18 months were 85.71% (95% CI: 79.86–89.97), 73.54% (95% CI: 66.64–789.24), and 55.56% (95% CI: 48.18–62.30), respectively.

### 3.3. Predictors Associated with 18-Month Mortality

Significant poor prognostic factors identified through univariable analysis included low BMI (<18.5), presence of contralateral lung metastasis, bone metastasis, low hemoglobin, high PMN, PLR ≤ 200, NLR ≥ 5, elevated Cr, AGR < 1, and MPA size ≥ 29 mm. Figure 2 illustrates the survival differences among patients with and without each of these prognostic factors. All significant variables were included in the full model, and Schoenfeld’s residual test indicated no violations of a proportional hazard assumption (*p* = 0.677).

After applying backward elimination, the final reduced model identified the significant predictors including BMI < 18.5, BMI ≥ 23, bone metastasis, NLR ≥ 5, AGR < 1, and MPA size ≥ 29 mm (Table 3). Beta-coefficients based on the log hazard scale and their 95% CIs for all predictors in both full and reduced models are detailed in Appendix A. The results were consistent with complete case analysis, showing minor variations in hazard ratios and 95% CIs (Appendix A).

### 3.4. Model Discrimination and Calibration

For model discrimination, the Harrell’s C-statistics for the full model and reduced model were 0.74 (95% CI: 0.69–0.79) and 0.72 (95% CI: 0.66–0.78), respectively. Somers’ D was 0.48 for the full model and 0.43 for the reduced model. Royston and Sauerbrei R^2^_D_ and D were 0.40 (95% CI: 0.28–0.51) and 1.687 (SE 0.214) for the full model, while the R^2^_D_ and D for the reduced model were 0.37 (95% CI: 0.24–0.48) and 1.555 (SE 0.214), respectively. For the calibration plot, the prognostic model was well calibrated in the second and fourth quantiles while the first and third showed partial underestimation and overestimation of the probability of death at the later survival time, respectively (Figure 3).

### 3.5. Internal Validation

After internal validation, the apparent, test, and optimism-adjusted values for Harrell’s C-statistic were 0.75 (95% CI: 0.74–0.75; Min 0.68; Max 0.83), 0.71 (95% CI: 0.71–0.71; Min 0.60; Max 0.74), and 0.04 (95% CI: 0.03–0.04), respectively. The optimism from Somers’ D, Royston R^2^_D_, and Royston D was 0.08 (95% CI: 0.07–0.09), 0.12 (95% CI: 0.11–0.13), and 0.41 (95% CI: 0.37–0.46), respectively. The shrinkage factor was estimated at 0.970. (Appendix A). The performance measures, including 95% CIs and ranges, showed high concordance with the complete case analysis, with only minimal variations (Appendix A).

### 3.6. Demonstration of Individual Predictions from the Model

From our dataset, we selected nine patients with different combinations of predictors to illustrate survival probabilities after 6, 12, and 18 months (Table 4). Each patient had a unique combination of variables, resulting in varied survival estimates. The survival prediction curves for each patient are shown in Figure 4. Patients without any poor prognostic factors have an estimated 18-month survival rate of 87.09% (95% CI: 78.68–92.34). In contrast, those with all poor prognostic factors have an estimated 18-month survival rate of only 2.13% (95% CI: 0.60–13.85). The results were consistent in the complete case analysis (Appendix A).

To enhance the applicability of the model, the beta coefficients from each predictor and baseline survival rates were extracted from the final model and incorporated into equations to calculate survival probabilities. These equations were implemented in a web application, which is publicly accessible at https://lung-ca-survival-prediction.streamlit.app/ (accessed on 1 August 2024).

## 4. Discussion

Nowadays, molecular-oriented treatment for advanced NSCLC is essential not only to enhance treatment response but also to extend patients’ survival [20]. EGFR-mutated NSCLC is highly prevalent in Thailand; however, access to EGFR-TKIs remains a significant challenge. Our study demonstrated an OS rate of 73.54% after 12 months and 55.56% after 18 months among patients treated with upfront EGFR-TKIs therapy. The majority (91%) of patients in this study received first-generation EGFR-TKIs, which aligns with findings from a multicenter retrospective cohort study in Thailand, where the OS of patients receiving EGFR-TKIs was reported to be 19.8 months [8]. Several landmark trials have shown that the median OS for first-generation EGFR-TKIs ranged from 22.8 to 36 months [7,21,22]. However, in this study, the median OS has not yet been reached.

Several studies have identified various prognostic factors influencing outcomes in advanced NSCLC [23]. To our knowledge, both cancer-related and host factors play a crucial role in determining the survival of NSCLC patients [24]. This study identified several poor prognostic factors that were independently associated with death at 18 months in patients with advanced EGFR-mutated NSCLC treated with EGFR-TKI, based on univariable analysis. These factors included low BMI group, contralateral lung metastasis, bone metastasis, low hemoglobin, high PMN, PLR ≤ 200, NLR ≥ 5, high Cr, AGR < 1, and MPA size ≥ 29 mm. After backward elimination, the prediction model was created using five prognostic factors as follows: BMI group, presence of bone metastasis, NLR ≥ 5, AGR < 1, MPA size ≥ 29 mm.

Using backward elimination, a prediction model was developed with five key prognostic factors: BMI, presence of bone metastasis, NLR ≥ 5, AGR < 1, and MPA size ≥ 29 mm. Clinically, BMI was a significant predictor of survival. Patients with a BMI < 18.5 had a significantly worse prognosis (HR 1.34), whereas those with a BMI ≥ 23 had a more favorable outcome (HR 0.52). This finding is consistent with the study by Evcil FY et al., which reported that underweight patients with advanced EGFR-mutated NSCLC had 2.08-fold (95% CI: 1.05–4.14) higher risk of death compared to those with normal weight [25]. Similarly, a pooled analysis of 29,217 lung cancer patients (35% with advanced disease) found that underweight patients at diagnosis had worse OS than those with normal weight (HR 1.56, 95% CI: 1.43–1.70) [26].

The presence of bone metastasis is a well-established indicator of poor prognosis in patients with various type of cancer, including lung cancer. In lung cancer patients with bone metastasis, the median OS has been reported as approximately 6–7 months, with the 1-year survival rate ranging from 12% to 30% [27,28]. Consistently, this study identified bone metastasis as a significant factor associated with reduced OS, with an HR of 2.08. A study on a Chinese cohort of patients with bone metastasis NSCLC reported poor OS, with an HR of 1.38, which was associated with poor PFS outcomes [10]. Moreover, the presence of multiple-bone metastasis was significantly associated with shorter OS compared to single-bone metastasis [29]. In contrast, other metastasis sites and mutational subtype were not identified as significant prognostic factors in this study. Regarding laboratory parameters, high NLR (NLR ≥ 5) was associated with poor prognosis, aligning with previous studies [30,31]. Additionally, a low AGR (AGR < 1) was also linked to worse prognosis. This finding was consistent with a meta-analysis showing that a low pre-treatment AGR was significantly associated with poor OS in lung cancer patients (HR = 1.88, 95% CI: 1.49–2.38, *p* < 0.001) [32]. Two retrospective studies further supported the prognostic value of AGR. One study from Turkey reported that in advanced adenocarcinoma of the lung, an AGR < 1.01 prior to chemotherapy was a significant predictor of long-term mortality [33]. Another study from China demonstrated that low AGR was significantly associated with poor OS and PFS [10]. Notably, even in early-stage NSCLC, an AGR less than 1.51 was identified as an independent prognostic factor for OS (HR = 3.424 (95% CI: 1.600–7.331, *p* = 0.002) [34]. Given these findings, both AGR and NLR may serve as valuable surrogate biomarkers for predicting mortality. Furthermore, an increase in the serum albumin level and a reduction in NLR following treatment may indicate improved survival outcomes in patients with advanced NSCLC [35].

The identification of relevant parameters from CT imaging is relatively straightforward, given its routine use in staging and monitoring treatment response in lung cancer patients. Among these parameters, the size of the MPA is particularly significant, as chronic pulmonary conditions such as COPD, especially prevalent among smokers, are recognized risk factors for lung cancer [14]. Although lung cancers with EGFR mutation are generally less associated with smoking, this study found that an MPA size ≥ 29 mm was significantly associated with increased 18-month mortality (HR 2.74). This cutoff value was selected based on its high sensitivity in identifying pulmonary hypertension confirmed by right heart catheterization (RHC) [36]. Furthermore, both PA enlargement and elevated pulmonary artery systolic pressure (PASP) were associated with reduced OS in lung cancer patients (HR 1.705, 95% CI: 1.296–2.243). According to that report, the primary predictors of elevated PASP were advanced age, coagulation disorder, and intra-pulmonary metastasis [37]. However, this study found no difference in age and intra-pulmonary metastasis between dead and surviving groups.

The National Cancer Institute of Canada Clinical Trials Group Study BR.21 identified ten factors associated with poor OS in EGFR-mutated NSCLC patients treated with erlotinib [38]. These included smoking history, performance status, weight loss, anemia, elevated lactate dehydrogenase levels, response to prior chemotherapy, time from diagnosis, number of prior treatment regimens, EGFR copy number, and ethnicity. However, the study population was heterogeneous, and EGFR-TKI was administered as a second- or later-line therapy, which does not align with current clinical practice, wherein EGFR-TKIs are recommended as first-line treatment in patients with this activating mutation.

A prognostic model developed by Du et al. [38,39,40] in a Chinese cohort incorporated ECOG performance status, EGFR mutation subtype, EGFR co-mutations, presence of liver metastasis, and malignant pleural effusion to predict PFS. However, the prognostic factors identified in their study differed from those in ours. Notably, our model demonstrated a higher C-statistic, indicating superior predictive performance compared to the previously proposed model. Additionally, a Korean cohort study developed a nomogram for predicting PFS in EGFR-mutated NSCLC [38,40]. The model included five factors: de novo metastatic disease at presentation, response to EGFR-TKI, ECOG performance status, and treatment line of EGFR-TKI. While the model demonstrated good discriminative ability with a C-index of 0.708, its applicability may be limited by the inclusion of patients receiving EGFR-TKIs in various treatment lines (first to third line), potentially diminishing its relevance to current first-line treatment settings.

To date, several studies have evaluated the prognostic factors and developed predictive models to enhance accuracy and reliability across various subtypes of lung cancer [41,42]. Building on this foundation, the present study introduced a robust prognostic model based on a readily available clinical variables, including pretreatment characteristics, laboratory results, and chest CT parameters. Our model integrated five key predictive factors and demonstrated strong predictive performance, with satisfactory discriminative power and calibration. Notably, it offered practical and reliable tool for predicting 18-month survival in patients with EGFR-mutated NSCLC undergoing EGFR-TKI therapy.

### Limitation

This study was a single-center retrospective cohort analysis. Nonetheless, the data were obtained from a large, well-characterized lung cancer cohort at a university medical center, with high-quality clinical, laboratory, and imaging data contributing to this study’s reliability. While third-generation EGFR-TKIs are now the global standard for first-line treatment due to their superior overall survival and progression-free survival outcomes, the majority of patients in our cohort received first-generation EGFR-TKIs, primarily due to reimbursement constraints. Consequently, the model’s predictive performance may not fully reflect outcomes in patients treated with newer-generation agents. Future study should externally validate this prognostic model in the larger, multicenter cohort, particularly among patients receiving third-generation EGFR-TKIs as first-line therapy.

## 5. Conclusions

The prognostic model developed from fundamental clinical and radiologic parameters demonstrated promising utility in predicting 18-month mortality in patients with advanced EGFR-mutated NSCLC receiving first-line EGFR-TKI therapy. By providing individualized risk stratification based on accessible pre-treatment data, this model could support clinicians in optimizing patient assessment, guiding treatment decisions, and facilitating informed prognostic discussions. Its integration into clinical practice has the potential to enhance personalized care and ultimately improve patient outcomes.

## Figures and Tables

**Figure 1 cancers-17-02565-f001:**
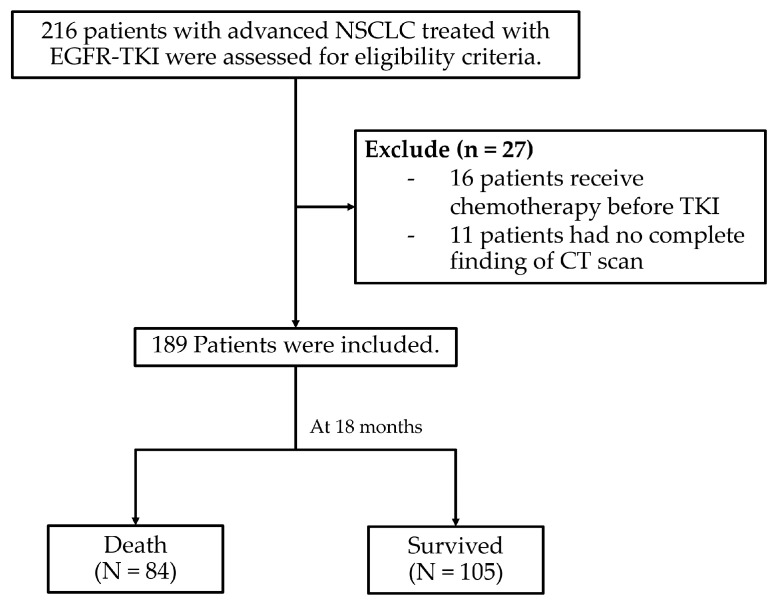
Study flow diagram.

**Figure 2 cancers-17-02565-f002:**
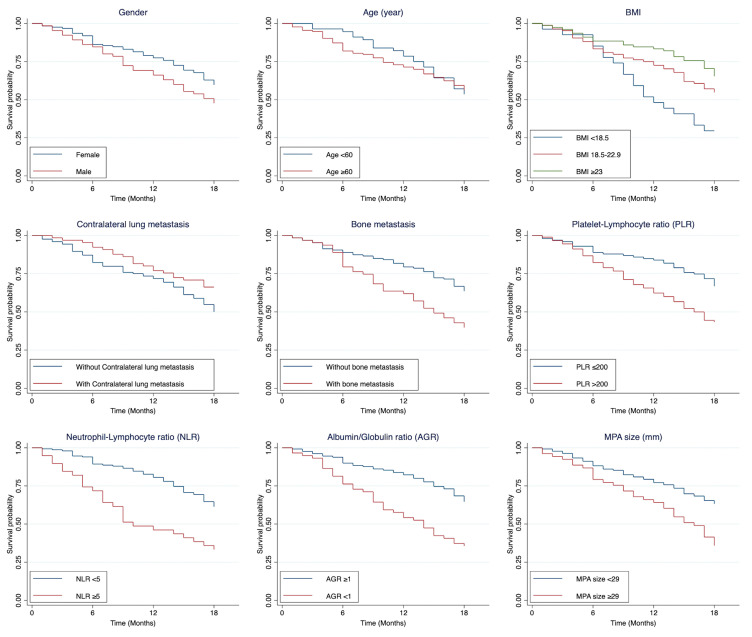
Kaplan–Meier curves visualizing differences in survival distribution among patients with and without prognostic factors. AGR, albumin-to-globulin ratio; MPA, main pulmonary artery (mm); NLR, neutrophil-to-lymphocyte ratio; PLR, platelet-to-lymphocyte ratio.

**Figure 3 cancers-17-02565-f003:**
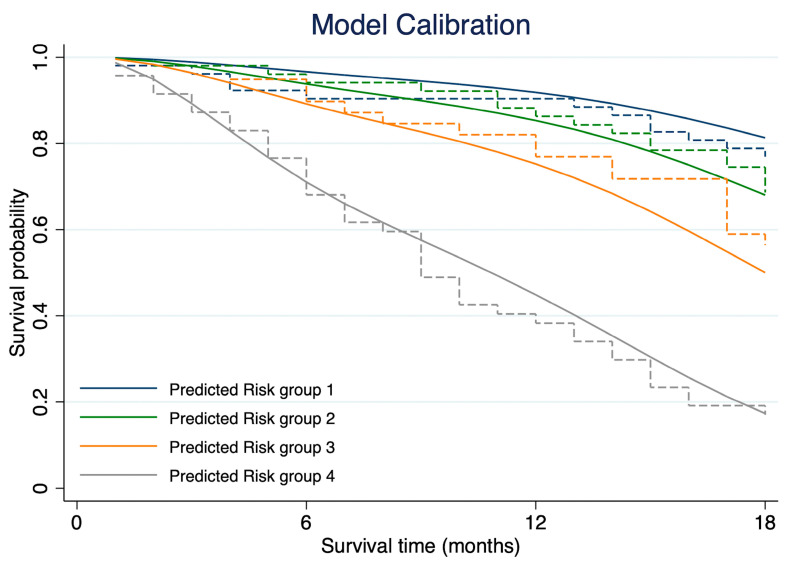
Calibration plots compare the model-predicted risk and the observed outcomes against one another within each of the risk quantiles.

**Figure 4 cancers-17-02565-f004:**
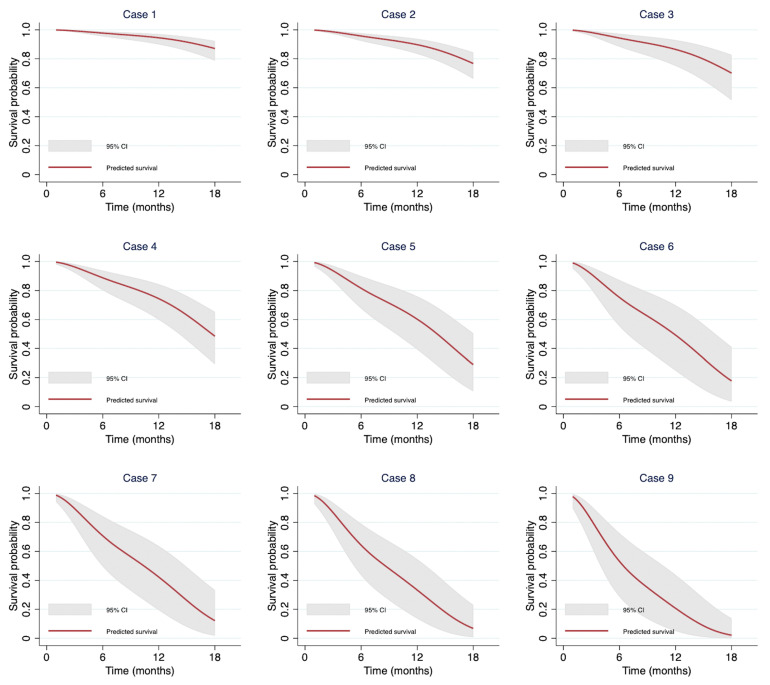
Individual prediction curves of nine randomly selected patients from the study cohort.

**Table 1 cancers-17-02565-t001:** Clinical characteristics of patient cohort.

Clinical Characteristics	Results	Missing N, (%)	Hazard Ratio * (95% CI)	6-Month Survival, % (95% CI)	12-Month Survival, % (95% CI)	18-Month Survival, % (95% CI)	*p*-Value **
Age (years), mean (SD)	64.81 (10.76)	0	1.00 (0.98–1.02)	-	-	-	0.809
Age group, N (%)		0					
Age < 60 years	56 (29.63)	-	Reference	94.64 (84.30–98.24)	78.57 (65.36–87.22)	53.57 (39.75–65.55)	0.997
Age ≥ 60 years	133 (70.37)	-	1.00 (0.63–1.59)	81.95 (74.30–87.52)	71.43 (62.93–78.31)	56.39 (47.54–64.32)	
Male, N (%)	65 (34.39)	0	1.46 (0.95–2.26)	84.62 (73.29–91.41)	66.15 (53.29–76.24)	47.69 (35.20–59.16)	0.080
BMI (kg/m^2^), mean (SD)	22.38 (4.05)	0	0.90 (0.85–0.96)	-	-	-	0.001
BMI group, N (%)		0					
<18.5	27 (14.29)	-	1.95 (1.12–3.39)	85.19 (65.20–94.17)	48.15 (28.69–65.19)	29.63 (14.06–47.03)	0.001
18.5–22.9	84 (44.44)	-	Reference	83.33 (73.49–89.77)	72.62 (61.74–80.88)	54.76 (43.54–64.66)	
≥23	78 (41.27)	-	0.69 (0.42–1.12)	88.46 (79.00–93.82)	83.33 (73.04–89.96)	65.38 (53.72–74.79)	
ECOG-PS, N (%)		0					
0–1	183 (96.83)	-	Reference	85.79 (79.84–90.10)	74.32 (67.34–80.03)	55.74 (48.24–62.58)	0.641
2–3	6 (3.17)		1.31 (0.41–4.15)	83.33 (27.31–97.47)	50.00 (11.09–80.37)	50.00 (11.09–80.37)	
Smoking, N (%)		0					
Never	119 (62.96)	-	Reference	86.55 (78.99–91.54)	75.63 (66.87–82.38)	58.82 (49.44–67.05)	0.196
Yes	70 (37.04)	-	1.32 (0.86–2.04)	84.29 (73.42–90.97)	70.00 (57.79–79.29)	50.00 (37.84–61.00)	
Comorbid disease, N (%)	107 (56.61)	0	0.72 (0.47–1.11)	83.18 (74.64–89.05)	77.57 (68.43–84.36)	60.75 (50.84–69.27)	0.131
COPD	6 (3.17)	-	1.86 (0.68–5.10)	66.67 (19.46–90.44)	66.67 (19.46–90.44)	33.33 (4.61–67.56)	0.208
Hypertension	82 (43.39)	-	0.74 (0.47–1.15)	82.93 (72.88–89.51)	76.83 (66.11–84.54)	62.20 (50.79–71.69)	0.169
Diabetes Mellitus	31 (16.40)	-	1.06 (0.60–1.89)	80.65 (61.91–90.80)	70.97 (51.62–83.71)	54.84 (35.97–70.26)	0.833
Other cancers	9 (4.76)	-	0.41 (0.10–1.67)	88.89 (43.30–98.36)	88.89 (43.30–98.36)	77.78 (36.48–93.93)	0.191
TKI, N (%)		0					
Gefitinib	91 (48.15)	-	Reference	84.62 (75.41–90.59)	73.63 (63.29–81.47)	54.95 (44.17–64.47)	0.540
Erlotinib	81 (42.86)	-	0.99 (0.63–1.54)	85.19 (75.39–91.30)	70.37 (59.14–79.05)	54.32 (42.89–64.41)	
Afatinib	7 (3.70)	-	1.10 (0.39–3.06)	100 (NA)	85.71 (33.41–97.86)	42.86 (9.78–73.44)	
Osimertinib	10 (5.29)	-	0.37 (0.09–1.54)	90.00 (47.30–98.53)	90.00 (47.30–98.53)	80.00 (40.87–94.59)	
Characteristics of cancer							
Staging of NSCLC, N (%)		0					
Locally advanced	3 (1.59)	-	Reference	100 (NA)	100 (NA)	66.67 (5.41–94.52)	0.569
Advanced	186 (98.41)	-	1.74 (0.24–12.53)	85.48 (79.55–89.81)	73.12 (66.13–78.90)	55.38 (47.94–62.18)	
Site of metastasis, N (%)		0					
Multiple sites	62 (32.80)	-	0.93 (0.59–1.47)	90.32 (79.72–95.53)	74.19 (61.38–83.32)	56.45 (43.25–67.70)	0.761
Contralateral lung	65 (34.39)	-	0.61 (0.37–1.00)	92.31 (82.50–96.72)	76.92 (64.67–85.39)	66.15 (53.29–76.24)	0.044
Brain	33 (17.46)	-	1.31 (0.77–2.23)	78.79 (60.59–89.27)	66.67 (47.94–79.96)	48.48 (30.83–64.06)	0.308
Bone	63 (33.33)	-	2.03 (1.32–3.12)	79.37 (67.13–87.46)	61.90 (48.76–72.60)	39.68 (27.67–51.43)	<0.001
Liver	17 (8.99)	-	0.55 (0.22–1.35)	100 (NA)	94.12 (65.02–99.15)	70.59 (43.15–86.56)	0.175
Adrenal	9 (4.76)	-	1.54 (0.67–3.54)	88.89 (43.30–98.36)	66.67 (28.17–87.83)	33.33 (7.83–62.26)	0.295
Pleural	82 (43.39)	-	0.98 (0.63–1.51)	85.37 (75.67–91.41)	73.17 (62.18–81.44)	56.10 (44.71–66.03)	0.918
Pericardial	4 (2.12)	-	1.81 (0.57–5.75)	100 (NA)	75.00 (12.79–96.05)	25.00 (0.89–66.53)	0.299

Abbreviations: CI, confidence interval; ECOG-PS, Eastern Cooperative Oncology Group performance status; NA, not available; NSCLC, non-small cell lung cancer; SD, standard deviation. * Hazard ratio from univariable Cox’s proportional hazard regression. ** *p* value was obtained from log-rank test for categorical variables, while continuous variables used *p*-value from Cox’s proportional hazard regression.

**Table 2 cancers-17-02565-t002:** Laboratory investigation and computer tomography (CT) findings.

Laboratory Characteristics	Results	Missing, N (%)	Hazard Ratio * (95% CI)	6-Month Survival, % (95% CI)	12-Month Survival, % (95% CI)	18-Month Survival, % (95% CI)	*p*-Value **
Hemoglobin (g/dL), mean (SD)	12.01 (1.76)	2 (1.06)	0.78 (0.69–0.90)	-	-	-	<0.001
WBC (cells/mm^3^), mean (SD)	8099.68 (2722.25)	2 (1.06)	1.00 (1.00–1.00)	-	-	-	0.548
PMN (%), mean (SD)	66.84 (10.70)	2 (1.06)	1.04 (1.02–1.06)	-	-	-	<0.001
Lymphocyte (%), mean (SD)	23.43 (11.21)	2 (1.06)	0.95 (0.93–0.98)	-	-	-	<0.001
PLT (×10^3^ cells/mm^3^), mean (SD)	329.91 (112.31)	2 (1.06)	1.00 (1.00–1.00)	-	-	-	0.166
PLR (plt/lymph ratio), med (IQR)	194.92 (135.14–279.14)	2 (1.06)	1.00 (1.00–1.00)	-	-	-	<0.001
PLR group, N (%)		2 (1.06)					
PLR ≤ 200	97 (51.87)	-	Reference	89.69 (81.69–94.32)	84.54 (75.67–90.37)	67.01 (56.70–75.39)	<0.001
PLR > 200	90 (48.13)	-	2.14 (1.37–3.33)	82.22 (72.63–88.71)	62.22 (51.37–71.32)	43.33 (32.95–53.24)	
NLR (neu/lymph ratio), med (IQR)	3.05 (2.00–4.62)	2 (1.06)	1.07 (1.03–1.10)	-	-	-	<0.001
NLR group, N (%)		2 (1.06)					
NLR < 5	148 (79.14)	-	Reference	89.86 (83.75–93.76)	81.08 (73.79–86.53)	61.49 (53.15–68.78)	<0.001
NLR ≥ 5	39 (20.86)	-	2.55 (1.60–4.06)	71.79 (54.88–83.28)	46.15 (30.16–60.73)	33.33 (19.29–48.02)	
Bun (mg/dL), med (IQR)	11.00 (9–14)	54 (28.57)	0.97 (0.92–1.02)	-	-	-	0.175
Cr, med (IQR)	0.75 (0.62–0.91)	4 (2.12)	0.27 (0.10–0.75)	-	-	-	0.012
BUN/Alb ratio, med (IQR)	2.41 (0–3.40)	10 (5.29)	1.05 (0.94–1.17)	-	-	-	0.429
Albumin, mean (SD)	3.92 (1.68)	10 (5.29)	0.34 (0.24–0.48)	-	-	-	<0.001
AGR (alb/glob ratio), mean (SD)	1.13 (0.44)	16 (8.47)	0.16 (0.07–0.41)	-	-	-	<0.001
AGR group, N (%)		16 (8.42)					
AGR ≥ 1	118 (68.21)	-	Reference	91.53 (84.82–95.35)	83.045 (74.97–88.71)	67.80 (58.55–75.41)	<0.001
AGR < 1	55 (31.79)	-	2.57 (1.62–4.10)	78.18 (64.79–86.97)	54.55 (40.56–66.56)	38.18 (25.53–50.71)	
CT finding characteristics							
Emphysema	20 (10.58)	0	1.36 (0.72–2.56)	90.00 (65.60–97.40)	70.00 (45.05–85.25)	45.00 (23.11–64.71)	0.334
MPA size (mm), mean (SD)	26.03 (3.93)	0	1.09 (1.04–1.15)	-	-	-	0.001
MPA group, N (%)		0					
MPA < 29 (mm)	137 (72.11)	-	Reference	88.24 (81.52–92.62)	77.21 (69.20–83.38)	63.24 (54.54–70.71)	<0.001
MPA ≥ 29 (mm)	53 (28.04)	-	2.08 (1.34–3.21)	79.25 (65.66–87.93)	64.15 (49.73–75.42)	35.85 (6.59–48.57)	
AO size (mm), mean (SD)	31.05 (4.10)	0	1.05 (1.00–1.11)	-	-	-	0.054
MPA to AO ratio, mean (SD)	0.84 (0.12)	0	3.63 (0.71–18.54)	-	-	-	0.121

Abbreviations: Alb, albumin; AO, aorta; CI, confidence interval; Glob, globulin; IQR, interquartile range; lymph, lymphocyte; Med, median; mm, millimeter; MPA, main pulmonary artery; neu, neutrophil; PLT, platelet; PMN, polymorphonuclear neutrophils; SD, standard deviation; WBC, white blood cell. * Hazard ratio from univariable Cox’s proportional hazard regression. ** *p* value was obtained from log-rank test for categorical variables, while continuous variables used p-value from Cox’s proportional hazard regression.

**Table 3 cancers-17-02565-t003:** Estimated hazard ratios in the full and reduced multivariable flexible parametric regression models.

Predictors	Full Model			Reduced Model		
	HR	95% CI	*p* Value	HR	95% CI	*p* Value
BMI group						
<18.5	1.35	0.75–2.45	0.320	1.34	0.76–2.38	0.313
18.5–22.9	1.00	Reference	NA	1.00	Reference	NA
≥23	0.58	0.34–0.99	0.045	0.52	0.31–0.88	0.015
Contralateral lung metastasis						
No	1.00	Reference	NA	Not include		
Yes	0.63	0.38–1.40	0.071			
Bone metastasis						
No	1.00	Reference	NA	1.00	Reference	NA
Yes	1.93	1.21–3.05	0.005	2.08	1.34–3.25	0.001
Hemoglobin (g/dL)	0.93	0.80–1.07	0.310	Not include		
PMN (%)	1.01	0.98–1.04	0.535	Not include		
PLR group						
PLR ≤ 200	1.00	Reference	NA	Not include		
PLR > 200	1.03	0.61–1.73	0.908			
NLR group						
NLR < 5	1.00	Reference	NA	1.00	Reference	NA
NLR ≥ 5	1.66	0.83–3.31	0.148	2.25	1.39–3.64	0.001
Cr	0.59	0.24–1.40	0.230	Not include		
AGR group						
AGR ≥ 1	1.00	Reference	NA	1.00	Reference	NA
AGR < 1	1.96	1.21–3.16	0.006	2.17	1.40–3.35	<0.001
MPA group						
MPA < 29 (mm)	1.00	Reference	NA	1.00	Reference	NA
MPA ≥ 29 (mm)	2.57	1.60–4.14	<0.001	2.74	1.72–4.36	<0.001

Abbreviations: AGR, albumin-to-globulin ratio; CI, confidence interval; HR, hazard ratio; mm, millimeter; MPA, main pulmonary artery; NLR, neutrophil-to-lymphocyte ratio; PLR, platelet-to-lymphocyte ratio; PMN, polymorphonuclear neutrophils.

**Table 4 cancers-17-02565-t004:** Demonstration of the model-estimated survival probability at each time point from nine sample patient.

Input Predictors						Model Estimation of Survival Probability (%, 95% CI)		
No.	BMI Groups	Bone Metastasis	NLR > 5	AGR < 1	MPA ≥ 29	6 Months	12 Months	18 Months
1	≥23	No	No	No	No	97.76 (95.43–98.90)	94.51 (89.90–97.05)	87.09 (78.68–92.34)
2	18.5–22.9	No	No	No	No	95.75 (92.36–97.65)	89.75 (83.37–93.77)	76.76 (66.07–84.47)
3	<18.5	No	No	No	No	94.34 (88.30–97.31)	86.49 (75.07–92.92)	70.11 (51.35–82.77)
4	18.5–22.9	No	No	No	Yes	88.78 (80.23–93.77)	74.35 (59.78–84.30)	48.43 (29.17–65.27)
5	18.5–22.9	Yes	Yes	No	No	81.57 (67.55–89.96)	60.22 (39.25–75.96)	28.92 (10.49–50.52)
6	<18.5	No	Yes	Yes	No	75.24 (55.34–87.22)	49.26 (24.76–69.82)	17.69 (3.43–41.06)
7	<18.5	No	No	Yes	Yes	70.74 (49.57–84.30)	42.23 (19.32–63.64)	12.14 (1.75–33.35)
8	18.5–22.9	Yes	Yes	Yes	No	64.31 (43.53–79.12)	33.32 (13.60–54.59)	6.80 (0.72–23.10)
9	≥23	Yes	Yes	Yes	Yes	53.15 (28.66–72.63)	20.73 (4.98–43.79)	2.13 (0.6–13.85)

Abbreviations: AGR, albumin-to-globulin ratio; CI, confidence interval; MPA, main pulmonary artery (mm); NLR, neutrophil-to-lymphocyte ratio.

## Data Availability

The data that support the findings of this study are available from the corresponding author, PT, pattraporn.t@cmu.ac.th, upon reasonable request. The data are not publicly available due to privacy or ethical restrictions.

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
