# Peer review of "Integrating Clinical and Imaging Markers for Survival Prediction in Advanced NSCLC Treated with EGFR-TKIs"

_cancers, 2025, doi:10.3390/cancers17152565_

Round 1

Reviewer 1 Report

Comments and Suggestions for Authors

The authors presented interesting research about clinical and imaging predictive markers for survival in advanced oncogene-addicted advanced NSCLC patients.

I would like to congratulate the authors for their hard work and consistent results.

Nevertheless, please find below my comments:

  1. First of all, please respect the type of article rules form Cancers and increase the size of summary, as a minim of 100 word were expected.
  2. Methods and Results are thoroughly described, but I don’t understand if variable parameters were assessed only at baseline or dynamically, during the study? I mean hematologic biomarkers, albumin, CT findings? Did you use RECIST criteria for EGFR-targeted therapy efficacy evaluation?
  3. Have you noticed the use of co-medication during the study? Meaning steroids, opioids, PPIs that have negative impact o immunotherapy response?
  4. In Discussion section the authors mainly presented one more the time the results of the study. Please revise this section and integrate each studied variable with the most recent published literature.

Eventually, this is a noteworthy piece of real-life evidence that diverse biomarkers might have predictive significance in NSCLC patients. Even if the authors drew attention about reimbursement restrictions and not using Osimertinib in first line, the baseline biomarkers and their evolution through cancer patients’ history (I point out here heavy burden disease, such as metastasis, low performance status, high NLR, low Albumin) are still negative prognostic and predictive factors. So, this is my opinion.

Author Response

Comment 1: First of all, please respect the type of article rules form Cancers and increase the size of summary, as a minim of 100 word were expected.

Answer: Thank you, revised version was done as your suggestion. (page 1, line 22-31)

Comment 2: Methods and Results are thoroughly described, but I don’t understand if variable parameters were assessed only at baseline or dynamically, during the study? I mean hematologic biomarkers, albumin, CT findings? Did you use RECIST criteria for EGFR-targeted therapy efficacy evaluation?

Answer:  Thank you for your comment. Regarding the prognostic model, all laboratory and CT finding variable parameters were assessed as a baseline before receiving systemic treatment. We clarified this point in page 4 line 162-163 “Prognostic factors including clinical characteristics, pre-treatment laboratory results and parameters associated with survival outcomes were explored.”

We used RECIST criteria for evaluate the response for EGFR-TKI treatment; however, the results were not reported, as this was not the primary objective of our study.

Comment 3: Have you noticed the use of co-medication during the study? Meaning steroids, opioids, PPIs that have negative impact o immunotherapy response?

Answer: Concurrent medication use was not systematically recorded in our study. However, the use of opioids was allowed during the treatment period. Proton pump inhibitors (PPIs), on the other hand, were restricted in the oncology clinic, and all patients received guidance and monitoring from pharmacists. Additionally, nine patients received a short course of corticosteroids while undergoing radiotherapy for brain or spinal metastases. Moreover, we included only patients receiving EGFR-TKI and none treated with immunotherapy.

Comment 4: In Discussion section the authors mainly presented one more the time the results of the study. Please revise this section and integrate each studied variable with the most recent published literature.

Answer: Thank you for point this aspect. We discussed more about other predictive models in the same setting in the discussion part. (page 16, line 393-412)

Reviewer 2 Report

Comments and Suggestions for Authors

The authors of the manuscript used a multivariate flexible parametric survival model to identify predictors of death at 18 months in patients with EGFR-mutant advanced NSCLC receiving first-line EGFR-TKI therapy. Predicted probabilities of survival at 6, 12, and 18 months were estimated, and the performance of the model was assessed. The estimated 18-month survival ranged from 87.1% in patients with no risk factors to 2.1% in patients with all predictors. A web-based tool was developed for clinical use.
Overall, I liked the article, there are a few minor comments:
1. Check the number of digits after the decimal point in the tables (Table 2), it should be brought to uniformity.

2. Did patients undergo surgical removal of the tumor before starting first-line EGFR-TKI therapy? If yes, this factor should also be included in the prognostic model.

3. The model also does not include the patient's condition according to the EGOG scale, the form of tumor growth (central/peripheral) and a number of other indicators. Why?

Author Response

Comment 1: Check the number of digits after the decimal point in the tables (Table 2), it should be brought to uniformity.

Answer: Thank you for your valuable feedback. We have made the corrections as you suggested. For the continuous data, we now present values with two decimal places. Additionally, we have revised the header labels to enhance clarity and understanding.

Comment 2: Did patients undergo surgical removal of the tumor before starting first-line EGFR-TKI therapy? If yes, this factor should also be included in the prognostic model.

Answer: Since our study focused on advanced disease, none of patients received surgery as a primary treatment prior to initiating EGFR-TKI treatment.

Comment 3: The model also does not include the patient's condition according to the EGOG scale, the form of tumor growth (central/peripheral) and a number of other indicators. Why?

Answer: ECOG performance status is presented in Table 1. The majority of patients (96.83%) had an ECOG score of 0–1 at the initiation of treatment; therefore, this variable was not included in the model due to limited variability. Additionally, the form of the tumor was not included in our analysis, as the study focused on evaluating non-cancerous factors—such as pulmonary artery size and emphysema—that may impact treatment outcomes.

Reviewer 3 Report

Comments and Suggestions for Authors

Predicting survival of NSCLC patients using imaging (particularly radiomics) and clinical data is a topic that has been widely explored over the last decade.

Any contribution could be valuable for further progress in this field and is therefore welcome.

Overall, the manuscript is clearly written, the English language is good (I suggest carefully checking for typos), and the background and methods are described in sufficient detail.

In particular, the manuscript focuses on survival prediction in patients with EGFR mutations treated with first-line EGFR-TKI therapy, using widely available and easily accessible clinical and biochemical data, as well as the most commonly performed radiological assessments, compared with more complex radiomic analyses, highlighting useful results in the specific geographical context analyzed.

I would suggest authors to start from this perspective to further improve their manuscript.

For example, it would be interesting to expand the bibliography by comparing data from Western countries with larger studies or by including different criteria and results.

Likewise, it would be interesting to compare the predictive results with those of other studies, for example using more complex parameters to calculate, ultimately highlighting the relevance of routinely collected and easily accessible medical data in clinical practice, especially in specific geographic or social contexts, as well as promoting progress in the availability of first-generation personalized care standards.

I would like to invite the authors to reflect on all the potential messages that could arise from their findings, to further strengthen the stated conclusions.

Author Response

Comment: 

Predicting survival of NSCLC patients using imaging (particularly radiomics) and clinical data is a topic that has been widely explored over the last decade.

Any contribution could be valuable for further progress in this field and is therefore welcome.

Overall, the manuscript is clearly written, the English language is good (I suggest carefully checking for typos), and the background and methods are described in sufficient detail.

In particular, the manuscript focuses on survival prediction in patients with EGFR mutations treated with first-line EGFR-TKI therapy, using widely available and easily accessible clinical and biochemical data, as well as the most commonly performed radiological assessments, compared with more complex radiomic analyses, highlighting useful results in the specific geographical context analyzed.

I would suggest authors to start from this perspective to further improve their manuscript.

For example, it would be interesting to expand the bibliography by comparing data from Western countries with larger studies or by including different criteria and results.

Likewise, it would be interesting to compare the predictive results with those of other studies, for example using more complex parameters to calculate, ultimately highlighting the relevance of routinely collected and easily accessible medical data in clinical practice, especially in specific geographic or social contexts, as well as promoting progress in the availability of first-generation personalized care standards.

I would like to invite the authors to reflect on all the potential messages that could arise from their findings, to further strengthen the stated conclusions.

Answer: We really appreciate your valuable comment. We added introduction on the comparison of our simple model with more complex models. Also with, other prediction models were summarized and discussed in the discussion part compared with our model. (page 3 line 103-107 and page 16 line 393-412)

Reviewer 4 Report

Comments and Suggestions for Authors

The manuscript attempts to develop a prognostic model for survival in EGFR-mutated advanced NSCLC patients receiving EGFR-TKIs. The study's focus on a clinically relevant issue is valuable, but it is hampered by several limitations.

  1. First-generation TKIs were administered to most of patients, while osimertinib was used in only 5%. This makes the model's practical use in current situations difficult.
  2.   Over the past three years, several validation cohorts have already combinedclinical+ CTradiomics+ blood-based biomarkers for EGFR-mutant NSCLC survival. The current model re-uses very similar candidate predictors  without any radiomics or deep-learning features, making the “new” contribution unclear.
  3.  Continuous variables were not analyzed using spline or fractional-polynomial methods for categorization. This discards prognostic granularity and artificially inflates HRs.
  4.  The C-statistic improved after bootstrapping, but the manuscript focuses on the uncorrected value in the abstract. This is misleading.
  5.  “thired-generation” should be “third-generation”.
  6.  28 % missing creatinine and 5 % missing albumin/globulin. k=5 kNN imputation was performed within VIM, but the authors provide no sensitivity analysis.

Author Response

Comment 1: First-generation TKIs were administered to most of patients, while osimertinib was used in only 5%. This makes the model's practical use in current situations difficult.

Answer: Thank you for your comment. Although, first-line osimertinib is widely used currently, the first generation EGFR-TKIs are mainly used among developing country including Thailand. The further research to validate our model among patients who receive first-line osimertinib would be benefit. This aspect was already discussed in the limiation part. (page 16, line 430-431)

Comment 2: Over the past three years, several validation cohorts have already combinedclinical+ CTradiomics+ blood-based biomarkers for EGFR-mutant NSCLC survival. The current model re-uses very similar candidate predictors  without any radiomics or deep-learning features, making the “new” contribution unclear.

Answer: Thank you for your comment. Our study focused on fundamental, easily accessible clinical parameters to develop a predictive model, rather than relying on complex and costly radiomic analyses. This approach was chosen to ensure the model’s applicability across a wide range of healthcare settings, including those with limited resources where advanced tools may not be available. Several parameters used in our model—such as hypoalbuminemia and low BMI—have previously been studied and shown to be associated with poor outcomes, but have not yet been integrated into a predictive model. Notably, our model is newly developed and is accompanied by a user-friendly, web-based tool designed to support practical use in routine clinical practice.

Comment 3: Continuous variables were not analyzed using spline or fractional-polynomial methods for categorization. This discards prognostic granularity and artificially inflates HRs.

Answer: We appreciate the comment regarding the handling of continuous variables. In our initial analysis, we categorized continuous variables to enhance clinical interpretability and facilitate their application in the final prediction model. The cutoff points were determined based on prior literature and established clinical practice. However, we acknowledge that categorization may limit prognostic granularity and could potentially inflate hazard ratios, as noted by the reviewer.

In our model development, we initially considered several continuous variables, including hemoglobin, absolute neutrophil count (used to calculate the neutrophil-to-lymphocyte ratio, NLR), and albumin (used in the albumin-to-globulin ratio, AGR). Nonetheless, both hemoglobin and creatinine did not demonstrate statistical significance in the multivariable model (Table 3) and were subsequently excluded during the backward elimination process.

Comment 4: The C-statistic improved after bootstrapping, but the manuscript focuses on the uncorrected value in the abstract. This is misleading.

Answer: We thank the reviewer for this important point. We acknowledge that presenting only the apparent (uncorrected) C-statistic in the abstract, without reporting the optimism-corrected value, could be misleading.

In response, we have revised the abstract to include both the apparent and optimism-corrected C-statistics, clearly indicating the value obtained after internal validation using bootstrapping (k = 500). The revised sentence now reads:

“The model demonstrated good performance (Harrell’s C-statistic = 0.72; 95% CI: 0.66–0.78). After internal validation, the apparent, test, and optimism for Harrell’s C-statistics, were 0.75 (95%CI 0.74 - 0.75), 0.71 (95%CI, 0.71 - 0.71) and 0.04 (95%CI, 0.03-0.04), respectively.”

Comment 5: “thired-generation” should be “third-generation”.

Answer: Thank you. We did correct this. (page 16, line 431)

Comment 6: 28 % missing creatinine and 5 % missing albumin/globulin. k=5 kNN imputation was performed within VIM, but the authors provide no sensitivity analysis.

Answer: We thank the reviewer for highlighting the importance of performing a sensitivity analysis to assess the robustness of our imputation approach. While we initially applied k-nearest neighbor (k=5) imputation using the VIM package, we recognize the importance of assessing the robustness of this choice.

To address this, we conducted a sensitivity analysis using a complete-case analysis. We compared key performance metrics of the prediction model between the imputed and complete-case datasets. The results showed consistent estimates, with a similar final reduced model and only minor variations in hazard ratios, C-statistics, key performance measures, and survival probability at each time point from nine sample patients (see supplementary Table S5–7). These findings support the robustness of our imputation approach and are now mentioned in the revised Methods and Results sections as follows:

Original version :

2.5. Missing data handling

Missing values in prognostic factors were addressed through imputation using the k-Nearest Neighbor (kNN) imputation based on a variation of the Gower Distance 17. Only prognostic factors were included in imputation procedures. The imputation was conducted in R using a kNN function from VIM package with default arguments setting at 5 neighbors, median function for numerical data, and maximum function for categorical data.

Revised version page 4, line 178-179:

2.5. Missing data handling

Missing values in prognostic factors were addressed through imputation using the k-Nearest Neighbor (kNN) imputation based on a variation of the Gower Distance 17. Only prognostic factors were included in imputation procedures. The imputation was conducted in R using a kNN function from VIM package with default arguments setting at 5 neighbors, median function for numerical data, and maximum function for categorical data. To assess the robustness of our imputation approach, we also performed a sensitivity analysis using a complete-case analysis.To assess the robustness of our imputation approach, we also performed a sensitivity analysis using a complete-case analysis.

Original version :

3.3. Predictors associated with 18-month mortality

Significant poor prognostic factors identified through univariable analysis included low BMI (<18.5), presence of contralateral lung metastasis, bone metastasis, low hemo-globin, high PMN, PLR ≤ 200, NLR ≥ 5, elevated Cr, AGR < 1, and MPA size ≥ 29 mm. Figure 2 illustrates the survival differences among patients with and without each of these prognostic factors. All significant variables were included in full model, and Schoenfeld’s residuals test indicated no violations of a proportional hazard assumption (P = 0.677).

After applying backward elimination, the final reduced model identified the sig-nificant predictors including BMI < 18.5, BMI > 23.5, bone metastasis, NLR ≥ 5, AGR < 1, and MPA size ≥ 29 mm (Table 3). Beta-coefficients based on log hazard scale and their 95% CIs for all predictors in both full and reduced models are detailed in Supplementary Table S3.

Revised version page 10 line 271-272:

3.3. Predictors associated with 18-month mortality

Significant poor prognostic factors identified through univariable analysis included low BMI (<18.5), presence of contralateral lung metastasis, bone metastasis, low hemo-globin, high PMN, PLR ≤ 200, NLR ≥ 5, elevated Cr, AGR < 1, and MPA size ≥ 29 mm. Figure 2 illustrates the survival differences among patients with and without each of these prognostic factors. All significant variables were included in full model, and Schoenfeld’s residuals test indicated no violations of a proportional hazard assumption (P = 0.677).

After applying backward elimination, the final reduced model identified the sig-nificant predictors including BMI < 18.5, BMI > 23.5, bone metastasis, NLR ≥ 5, AGR < 1, and MPA size ≥ 29 mm (Table 3). Beta-coefficients based on log hazard scale and their 95% CIs for all predictors in both full and reduced models are detailed in Supplementary Table S3. The results were consistent with complete case analysis, showing minor variations in hazard ratios and 95% CIs (supplementary table S4).

Original version :

3.5. Internal validation

After internal validation, the apparent, test, and optimism for Harrell’s C-statistics, were 0.75 (95%CI 0.74 - 0.75; Min 0.68; Max 0.83), 0.71 (95%CI, 0.71 - 0.71; Min 0.60; Max 0.74). and 0.04 (95%CI, 0.03-0.04), respectively. The optimism from Somers’ D, Royston R2D, and Royston D were 0.08 (95%CI, 0.07-0.09), 0.12 (95%CI 0.11-0.13), and 0.41 (95%CI 0.37-0.46), respectively. The shrinkage factor was estimated at 0.970(Supplementary Table S5).

Revised version page 12 line 301-303:

3.5. Internal validation

After internal validation, the apparent, test, and optimism for Harrell’s C-statistics, were 0.75 (95%CI 0.74 - 0.75; Min 0.68; Max 0.83), 0.71 (95%CI, 0.71 - 0.71; Min 0.60; Max 0.74). and 0.04 (95%CI, 0.03-0.04), respectively. The optimism from Somers’ D, Royston R2D, and Royston D were 0.08 (95%CI, 0.07-0.09), 0.12 (95%CI 0.11-0.13), and 0.41 (95%CI 0.37-0.46), respectively. The shrinkage factor was estimated at 0.970(Supplementary Table S5). The performance measures, including 95% CIs and ranges, showed high concordance with the complete-case analysis, with only minimal variations (Supplementary Table S6).

Original version :

3.6. Demonstration of individual predictions from the model

From our dataset, we selected nine patients with different combinations of predictors to illustrate survival probabilities at 6, 12, and 18 months (Table 4). Each patient had a unique combination of variables, resulting in varied survival estimates. The survival prediction curves for each patient are shown in Figure 4. Patients without any poor prognostic factors have an estimated 18-month survival rate of 87.09% (95% CI: 78.68–92.34). In contrast, those with all poor prognostic factors have an estimated 18-month survival rate of only 2.13% (95% CI: 0.60–13.85).

Revised version page 12 line 311-312:

3.6. Demonstration of individual predictions from the model

From our dataset, we selected nine patients with different combinations of predictors to illustrate survival probabilities at 6, 12, and 18 months (Table 4). Each patient had a unique combination of variables, resulting in varied survival estimates. The survival prediction curves for each patient are shown in Figure 4. Patients without any poor prognostic factors have an estimated 18-month survival rate of 87.09% (95% CI: 78.68–92.34). In contrast, those with all poor prognostic factors have an estimated 18-month survival rate of only 2.13% (95% CI: 0.60–13.85). The results were consistent in complete case analysis (supplementary table S7).

Furthermore, the supplementary materials were revised and tables were updates respectively, as showed in line 415-420.

Revised version page 17 line 443-448:

Supplementary Materials: The following supporting information can be downloaded at: https://www.mdpi.com/article/doi/s1, Table S1: Sample size calculation per group; Table S2: Mutational subtypes; Table S3: Estimated log hazard ratios in the full and reduced multivariable flexible parametric regression models; Table S4: Estimated hazard ratios in the full and reduced multivariable flexible parametric regression models using complete case analysis; Table S5: Performance measures from internal validation of 200 bootstrap samples; Table S6: Performance measures from internal validation of 200 bootstrap samples using complete case analysis; Table S7: Demonstration of the model-estimated survival probability at each time point from nine sample patient, estimated from complete case analysis

Round 2

Reviewer 4 Report

Comments and Suggestions for Authors

The authors have provided adequate responses to raised issues, and it is recommended for publication.